# Effectiveness, Acceptability and Feasibility of Technology-Enabled Health Interventions for Adolescents Living with HIV in Low- and Middle-Income Countries: A Systematic Review

**DOI:** 10.3390/ijerph20032464

**Published:** 2023-01-30

**Authors:** Talitha Crowley, Charné Petinger, Azia Ivo Nchendia, Brian van Wyk

**Affiliations:** 1School of Nursing, Faculty of Community and Health Sciences, University of the Western Cape, Cape Town 7535, South Africa; 2School of Public Health, Faculty of Community and Health Sciences, University of the Western Cape, Cape Town 7535, South Africa

**Keywords:** acceptability, adolescents, HIV, technology, low- and middle-income countries

## Abstract

Background: Adolescents living with HIV (ALHIV) are challenged to remain adherent and engaged in HIV care. Technology-enabled interventions can be used to optimize healthcare delivery to adolescents. The largest proportion of ALHIV resides in sub-Saharan Africa. This review synthesized the evidence for the effectiveness, acceptability, and feasibility of technology-enabled health interventions for ALHIV in low and middle-income countries (LMIC). Methods: Eight electronic databases (Ebscohost, CINAHL, ERIC, MEDLINE, PubMed, SCOPUS, Science Direct, and Sabinet) and Google Scholar were searched to identify studies in LMIC published from 2010 to 2022. Quantitative and qualitative studies reporting on technology-enabled health interventions for predominantly adolescents (10–19 years) were included. The review was performed, and findings were reported according to the Preferred Reporting Items for Systematic Reviews and Meta-analyses Protocols. The review was registered with PROSPERO: CRD42022336330. Results: There is weak evidence that technology-enabled health interventions for ALHIV in LMIC improve treatment outcomes. However, most interventions appear to be acceptable and feasible. Conclusion: There is a need to ensure that technology-enabled interventions have a strong theoretical base. Larger studies with rigorous evaluation designs are needed to determine the effects of these interventions on the health outcomes of ALHIV in LMIC.

## 1. Introduction

With the advent of and increased access globally to antiretroviral treatment (ART), human immune deficiency virus (HIV) can be well controlled in individuals who maintain high levels of treatment adherence and viral suppression. However, long-term adherence and engagement in care remain a concern, especially for adolescents living with HIV (ALHIV) [1,2]. Adherence is challenging due to the physiological, emotional/psychological, and social changes that take place within and around adolescence [3]. As ALHIV transition from pediatric to adult care, they are required to take on increased responsibility for medication adherence, clinic visits, and managing their general physical and mental wellbeing [4,5,6].

ALHIV is recognized as a key population group that requires unique interventions. The majority (80%) of the 1.75 million ALHIV (80%) globally reside in sub-Saharan Africa [7]. Living in low and middle-income countries (LMIC) presents a myriad of sociopolitical and contextual challenges such as poverty, violence, and fewer resources to provide differentiated healthcare to adolescents [8,9]. There is a paucity of evidence-based interventions for improving the outcomes of ALHIV in LMICs. Technology-enabled interventions have shown potential to improve outcomes in high-income countries such as the USA [10], but the evidence regarding their effectiveness, acceptability, and feasibility in LMIC is lacking.

Adolescents regularly use the internet or other social media platforms to search for health information [11,12]. As a result, there has been an increase in technology-enabled health interventions for adolescents. Technology-enabled health interventions are defined as interventions that use electronic devices, such as mobile phones or computers, for accessing information and communicating via the internet or a mobile network [13,14,15]. These interventions can be used to support ALHIV with self-management, promoting autonomy, adherence, goal-setting, and problem-solving [3,16], by providing privacy, support, and feedback [12].

Systematic reviews have been conducted on the effectiveness of technology-enabled interventions for health promotion [15,17,18,19,20], prevention [21,22], mental health [23,24] and chronic conditions [10,14] amongst adolescents. However, the applicability of these interventions across varied contexts, marginalized groups and different health conditions is limited [25]. Reviews on the effect of technology-enabled interventions amongst adults living with HIV have found that they can improve treatment access, symptom- and self-management, adherence, retention in care, risk reduction, quality of life and mobilize health care and social support [5,26,27,28]. In sub-Saharan Africa, e-health interventions for HIV prevention and management were found to be a low-cost way to improve adherence and retention in care [29].

There has been an increase in the number of technology-enabled interventions for ALHIV. A 2015 systematic review on technology-enabled interventions for ALHIV found only one study outside the USA [2]. A review of interventions to improve ART adherence (2016–2018) found one technology-enabled intervention (mHealth/SMS) [30] and a systematic review of self-management interventions for ALHIV (2000–2019) found four technology-enabled interventions (none in LMIC) [31]. More recently, a systematic review on m-health interventions for adolescents and young adults across the HIV prevention and continuum of care in LMIC (2000–2021) identified nine interventions for the delivery of care and treatment for ALHIV [32]. However, the review focused mainly on quantitative HIV treatment outcomes and not on feasibility, acceptability, and fidelity, which are key to understanding intervention implementation in particular contexts and for guiding the development of future interventions. It is necessary to identify design features as well as adolescent preferences for technology-enabled health interventions to guide future intervention development and scale-up. 

The current review aims to determine the effectiveness, feasibility, and acceptability of technology-enabled interventions in affecting health-related outcomes for ALHIV in LMIC. 

The review was guided by the following questions:What technology-enabled interventions have been implemented in LMIC to support and deliver healthcare to ALHIV (aged 10–19 years)?What is the effectiveness of various technology-enabled health interventions in general health and well-being and treatment outcomes of ALHIV in LMIC?What is the feasibility, acceptability, and fidelity of the various technology-enabled health interventions for ALHIV in LMICs?

## 2. Materials and Methods

### 2.1. Study Design 

The review was guided by the seven systematic review steps as described by Egger, Davey, and Smith [33]. The steps are to formulate the review question, determine the inclusion and exclusion criteria, develop the search criteria, select studies, assess the quality of the studies, extract the data, and analyze and synthesize the data. The study followed the guidelines provided in the Preferred Reporting Items for Systematic Reviews and Meta-analyses Protocols (PRISMA-P) [34] and was registered with the International Prospective Register of Systematic Reviews (PROSPERO) on 13 June 2022 (reference: CRD42022336330). 

### 2.2. Inclusion and Exclusion Criteria

We considered studies to be eligible for inclusion in the review if they met the following eligibility criteria.
Types of participants: Adolescents living with HIV between the ages of 10 and 19 years as the primary study population. We considered variations in age ranges if the adolescents were the predominant focus of the intervention, as it was difficult to find disaggregated data specific to adolescents aged 10–19 years.Types of interventions: Studies that describe a technology-enabled intervention to deliver or support healthcare (defined as interventions that use electronic devices such as mobile phones or computers for health information communication).Types of studies: Quantitative (randomized controlled trials, non-randomized controlled trials, before- and after studies) and qualitative studies reporting on the feasibility and acceptability of technology-enabled interventions. Peer-reviewed studies were published in the English language, conducted in LMIC, and published between 2010 and 2022.Types of comparisons: Technology-enabled health intervention vs. no intervention, the standard of care, waitlist, or another intervention with no technology-enabled component. We also considered studies with no comparison.Types of outcomes: We considered studies reporting on any health-related individual outcomes as defined by the study authors. We report on qualitative data related to the feasibility, acceptability, and fidelity of the intervention.

The PICOT (Population, Intervention, Comparison, Outcome and Time) criteria are summarized in Table 1. 

The exclusion criteria of this systematic review are as follows:Review studies.Technology-enabled interventions that do not involve the adolescent directly as a recipient of the intervention, i.e., electronic health registers, monitoring and recording of service delivery.

### 2.3. Information and Search Strategy

We used a broad search strategy to include technology-enabled health interventions in LMIC for adolescents. An information specialist was consulted to develop the search strings. The systematic search of databases was conducted on the following databases: Ebscohost (Psycharticles, Academic Search Premier), Cumulative Index of Nursing and Allied Health Literature (CINAHL), Educational Resource Information Center (ERIC), Medical Literature Analysis Retrieval System Online (MEDLINE), PubMed, SCOPUS, Science Direct, and Sabinet. The full-text articles were sourced by using the “AND” and “OR” Boolean operators and the following search terms/keywords and their MeSH terms. Box 1 contains the search strategy finalized in PubMed with an author and an information specialist, which was also used for the other databases.

Box 1PubMed Search Strategy.(“adolescent” OR “young people” OR “teen” OR “teenager”) AND(“Information and Communications Technology” OR “ICT” OR “Technology” OR “Technology Enabled” OR “Technology based” OR “gaming” OR “social media” OR “ehealth” OR “mhealth” OR “whatsapp” OR “SMS” OR “mobile” OR “internet” OR “text message” OR “telemedicine”) AND(“HIV” OR “AIDS”) AND(“Low-income countries” OR “Middle income countries”)

ClinicalTrials.gov (www.ClinicalTrials.gov, accessed on 15 November 2022) and the World Health Organization (WHO) trials portal (www.who.int/ictrp/en/, accessed on 15 November 2022) were searched to identify unpublished and ongoing studies. In addition, we searched grey literature such as university theses/dissertation databases and conference abstracts, for example, the International AIDS Conference, the Conference on Retroviruses and Opportunistic Infections (CROI) and the International Workshop on HIV and Adolescence. In addition, a search was conducted on Google Scholar. 

We screened reference lists of included studies and relevant systematic reviews to complement the electronic search.

### 2.4. Study Selection

For the database search the inclusion criteria and search strategy was used. The number of hits from each database was recorded and the citations were imported into Covidence software [35]. Covidence software allowed the removal of duplicates and the recording of the number of citations. Two reviewers (TC & CP) screened titles and abstracts for eligibility for inclusion. The full-text articles of eligible studies were retrieved following the title and abstract screening and were independently reviewed by two reviewers (CP and IA), for inclusion. Discrepancies were resolved through discussion and were resolved by the research team (TC, BvW, CP, and IA). The study selection process can be seen in the PRISMA diagram (see Figure 1). 

### 2.5. Data Extraction

Two authors (CP & IA) independently extracted data in Covidence using a pre-specified data extraction form. Prior to commencing data extraction, the form was piloted on one study identified for inclusion and modified accordingly. Data were extracted on the study design, characteristics of participants, type of intervention, description of the intervention, outcomes, and setting. A description of the components of the technology-enabled health interventions was extracted using an adapted form following the 12-item Template for Intervention Description and Replication (TIDier) checklist [36]. This assisted in recording important aspects of the intervention, such as the name of the intervention, aim/goal, theoretical foundation, involvement of end-users, duration and intensity, materials/content, procedure (including persons delivering the intervention), the type of device, technology design, and the delivery platform/mode. Disagreements were resolved through discussions with the entire research team. 

### 2.6. Risk of Bias and Quality Assessment

Quality assessment for all the studies was done using the Mixed Methods Appraisal Tool (MMAT) by Hong et al. [37]. The MMAT permitted the appraisal of the methodological quality of the five categories of studies used in the review: qualitative research, randomized controlled trials, non-randomized studies, quantitative descriptive studies, and mixed methods studies. Two reviewers (CP & IA) independently assessed the methodological quality of all included studies based on their mythological domain (see Table A2). Differences were discussed amongst the review team. The presence or absence of an assessment criterion was respectively given a score of 1 and 0 [38]. The final quality score for each article was found by dividing the total points scored by the article by the total points assessed for the article. The grading of the quality of the article was performed as follows: scores of ≤0.50 were rated as weak, 0.51 to 0.65 as moderately weak, 0.66 to 0.79 as moderately strong, and ≥0.80 as strong [39]. 

### 2.7. Data Synthesis

This review synthesized the results narratively. The authors tabulated and provided a narrative description of the summarized data. This was applicable to the review as the included articles were few and the data were heterogeneous. Further, it allowed for a clear description of the results, specifically the characteristics of the included articles, and their findings.

### 2.8. Patient and Public Involvement

As this is a systematic review, no patients or the public were involved in the design or research.

## 3. Results

### 3.1. Study Characteristics

Out of the 267 records identified, we screened 120 titles of studies based on the inclusion criteria. Of this number, eleven studies met the inclusion criteria and were included. Five ongoing studies were identified (see Table A1). The study selection process is provided in a PRISMA diagram (Figure 1).

A summary of the relevant characteristics of the included studies is provided (Table 2). 

A majority of the included studies took place in Africa. Three studies were conducted in Nigeria [40,43,44], two in Uganda [48,49], two in South Africa [45,46], and two in Kenya [41,42,47]. The remaining studies were conducted in South America, with a randomized control trial in Guatemala [50], and a non-RCT in Argentina [51].

Five of the studies [40,44,48,49,50] used an RCT design, while four other studies [41,42,43,47,51] used a non-RCT pre-post design. One study [45] used a non-RCT matched controls design and one study [46] used a qualitative design.

A total of 1544 participants were included in all the selected studies. Out of this number, 846 participants were females while 387 were males. Nonetheless, two studies [49,50] did not report on the number of female and male participants that were used in their studies. The age range of the participants in the studies was 6–25 years. In addition to age and ART eligibility criteria, four studies [43,44,47,48] required participants to demonstrate basic internet, SMS, or web-based literacy. Five studies [40,48,49,50,51] required participants to have a personal mobile phone or have access to mobile phones as an inclusion criterion. In one study [41,42], the participants were provided with a smartphone, the WhatsApp^®^ application preinstalled, a SIM card, and phone credit. 

### 3.2. Quality Assessment

Three RCTs [44,48,50] were each graded as moderately strong, and two as strong [40,49]. Concerning the grading of the non-RCT quantitative studies, three pre-post studies [41,42,47,51] were respectively graded as moderately strong and one [43] as weak. The matched-controlled study [44] was graded as strong. Although the study by Henwood et al. [46] was graded as strong, it was a qualitative study. See Table 3.

### 3.3. Characteristics of Technology-Enabled Interventions

In terms of the technological design of the interventions (see Table 4), the majority (five) involved interactive groups [41,42,43,44,46,47]. This means that these studies involved interactions between group members. Three studies used interactive individual designs [40,45,51] and two non-interactive individual designs [49,50]. Interactive and non-interactive individual designs involved SMS messages that either required the participant to respond or engage, or not. One study [48] compared interactive (message and response) vs. non-interactive (message only) vs. control. 

The delivery platforms were primarily social media and SMS-based. Four studies used social media (WhatsApp/Facebook/Mxit) to deliver the intervention [41,42,43,44,46], and four utilized SMSs as delivery [40,48,49,50]. For the remaining studies, one intervention was web-based [47], one mixed- using SMS, WhatsApp, and phone calls for delivery [45], and another provided the participants with the option to receive and respond to messages via phone or Facebook [51]. 

All interventions of the included studies involved end-users, and none specified whether a theoretical framework was used. 

In terms of the duration, the interventions were conducted between six and 18 months. Most of the studies took place for 12 months [40,47,48,51]. Two studies had a duration of six months [41,42,43] and one for nine months [50]. Five studies had a duration of longer than 12 months, namely two for 13 months [43,46], one for 14 months [45], and one for 18 months [49]. 

### 3.4. Assessment of the Effectiveness of Technology-Enabled Interventions: Primary Outcomes

Primary outcomes are summarized in Table 5. Of the eight studies that measured adherence as one of the primary outcomes [40,41,42,44,47,48,49,50,51] two studies [50,51] found a significant intervention-related improvement in ART adherence. In the study by Steinkievich et al. [51], viral load (VL) was measured as an indication of adherence. After 32 weeks of consecutive implementation of the intervention (generic text messages), 20 of 22 patients had VL measured in the context of a routine clinical visit. The limit of detection of the VL test was 40 copies/mL. Thirteen of 20 (65%) patients had an undetectable VL and 14 of 20 (70%) had VL < 1000 copies/mL while six out of 20 (30%) of the patients had no changes in the VL. Similarly, the study by Sánchez et al. [50] found that, from the study initiation to the final adherence measure, the text message intervention group demonstrated improved adherence (measured by a four-day recall questionnaire) by 4% (*p* < 0.01) while the control group experienced a non-significant adherence improvement of 0.85 percentage points (*p* = 0.64). 

No significant improvement or differences across groups were found in the other studies that assessed adherence. Within the studies, adherence was measured in different ways ranging from subjective measures, such as a visual analogue scale (VAS) and the AIDS Clinical Trials Group (ACTG) adherence questionnaire [40], to the Comprehensive ART measure for Pediatrics (CAMP) questionnaire [41,42], and other self-report/recall measures [47,50]. Some studies also used objective measures such as pill count, viral suppression [40,51], Medical Event Monitoring System (MEMS) capsules, or the Wisepill device [41,42,48,49]. 

Three studies [40,45,51] measured VL and only Steinkievich et al. [51] found a significant improvement in VL as discussed above using a cut-off of VL < 1000 copies/mL. Hacking et al. [45] used a cut-off of VL < 400 copies/mL for viral suppression and Abiodun et al. [40] used a value of <20 copies/mL.

One study, the Virtual Mentor’s Programme [45] assessed linkage to care and reported improvement in linkage to care measured by increased ART initiation in 28 of 35 (80%) individuals in the mentee group vs. 30 out of 70 (42%) in the matched controls. 

None of the studies that reported on retention in care [44,45] reported significant effects. Retention in care was measured as either not missing any appointments during a period, e.g., 28 days [44], or the number of participants retained in care after a period of six or 12 months [45].

One study showed a significant improvement in HIV knowledge [45], while another study did not know improvement [47]. Dulli et al. [45] used closed Facebook groups for online sessions and communication on a range of topics over six months. They found significantly better HIV-related knowledge (14 questions) in the intervention group at the end of the study (*p* = 0.003). Ivanova et al. [47] used a web-based digital peer support platform (12 months duration) and measured knowledge using 17 true/false items. They found an improvement in knowledge by 0.3 points, but it was not significant.

Studies that measured social support, self-efficacy, mental health, stigma, or behavioral outcomes [41,42,44,47] did not show any significant effect of the intervention on these outcomes.

### 3.5. Assessment of Secondary Outcomes

Table 6 provides a summary of the secondary outcomes. Except for the study by Linnemayr et al. [48], which did not report on secondary outcomes, all the studies had some measure of acceptability, feasibility, or fidelity through either self-report questionnaires on usability, willingness to continue the intervention, or qualitative interviews/focus groups.

We considered acceptability, feasibility, and fidelity to be high if more than 90% or the large majority of the study participants demonstrated acceptance of the intervention assessed. Ten of the eleven studies reviewed demonstrated high acceptability of the mobile technology used in the respective interventions. In two studies [46,49], participants reported that online groups should be complementary to face-to-face clinic visits. One study [50] found that acceptability of the SMS intervention was associated with reading literacy, cellphone ownership, reliable network, previous use of cellphone functions, and privacy of text messages.

Concerning feasibility, only one study [50] did not report on the feasibility aspect of the intervention, but all the other studies showed high feasibility of the respective information-communication technology used in the respective interventions. 

Regarding fidelity, three studies [43,44,49] showed high fidelity, and one study [46] low fidelity. Henwood et al. [46] used a virtual support group using Mxit and some participants did not participate due to forgetting the chat room password. Other fidelity challenges related to the device, network connectivity, and data challenges. One of the studies that used Facebook groups found that quizzes and polls did not appear correctly formatted and some struggled to upload photos of their adherence plans [43], indicating that phone capabilities should be considered. Fidelity appeared to be better in the online support groups if the facilitator was trained, reliable, and engaging and the participants felt comfortable with the facilitator [43,45]. 

Some participants of online groups were concerned about anonymity and confidentiality as they had no control over whether other participants shared content publicly [41,42,43,46]. Another challenge of the online groups was encouraging the participation of all members [43]. In the study that used virtual mentoring [46], participants commented that they preferred a more formal structure with topics [46]. There appeared to be a preference for the use of existing applications, e.g., WhatsApp or telephone calls in interventions that used Facebook or Mxit [43,44,46], because it also uses minimal data and chat histories are available should participants not be able to access the live chats [46].

Appropriate scheduling of online support groups or interactive SMS messages was identified as important as household or school responsibilities can be barriers to active participation. Some youth wanted to access groups and content at their leisure [41,42,46]. Caregiver engagement is important since some adolescents reported that caregivers did not approve of their phone use [41,42].

Qualitative data indicated the potential benefit of technology-enabled health interventions for ALHIV. Participants reported that the groups created a sense of hope, boosted morale, and provided a feeling of community and peer support among ALHIV that for many had not been previously available [41,42,49]. Further, some participants enjoyed the competition created through sharing adherence information. However, there were instances where the wrong information was shared due to technical difficulties which discouraged participants [49].

## 4. Discussion

Interventions in LMIC to support and deliver healthcare to ALHIV ranged from interactive groups on social media platforms to simple generic text messages. There appears to be a move towards interactive group or individual interventions, particularly to increase the social support of ALHIV. This makes sense due to the concerns of social isolation and mental health issues amongst ALHIV [1,2,3,4]. 

None of the interventions reported an underpinning theory that informed the study. Similarly, in an integrative review of US studies, less than half of the studies reported a theoretical framework [10]. It was, further, not clear whether sustainability and/or integration of the intervention into existing health care services were considered. International guidelines specify that technology-enabled interventions should use a theory-driven approach and that developers should consider how these interventions will complement existing health care or community services [52].

Two out of the eight interventions showed moderately strong evidence of effectiveness to improve adherence through either non-interactive SMS or interactive messaging, respectively. An integrative review of technology-enabled interventions for adherence support and retention in care among US ALHIV demonstrated the initial efficacy of SMS-texting for improving adherence in two studies and computer-based motivational interviewing for improving adherence in one study [10]. With regards to HIV knowledge, contradictory findings were reported and the effect on other outcomes, such as social support, stigma, mental health, and behavior, was non-significant. Our review did not find strong evidence of the effectiveness of technology-enabled health interventions on the health outcomes of ALHIV in LMIC. Other systematic reviews have supported the potential efficacy of mobile health interventions [29,32], but also concluded that most interventions were short-term and pilot studies with no evidence of the effect of scaled interventions. A further aspect that could be explored is the difference between interventions that involve humans and interactive components vs. those only utilizing technology [10]. However, there is not enough evidence currently to explore the effect of interventions and their acceptability across sub-groups of the type of technology design or sub-groups of the population, such as adolescents in different developmental stages. 

Technology-enabled interventions for ALHIV appear feasible and acceptable. Similarly, in the US, SMS and computer-based interventions demonstrated feasibility and acceptability [10]. Most interventions required ALHIV to have access to mobile phones. Access to mobile phones and the acceptability of phone use is increasing in LMICs [11,12], but intervention developers should consider device capability, internet connectivity and data access. Further, current interventions that use existing popular platforms such as SMS or WhatsApp appear highly acceptable and feasible and may have a better uptake compared to Facebook or web-based interventions. Online platforms have the advantage over simple SMS in that they can incorporate audio-visual components and interactive activities that engage adolescents [12]. One concern though is privacy and anonymity. Participants may have little control over whether other group members may share information publicly [43]. None of the interventions used specially developed smartphone applications or gamification features, although these are being developed in Nigeria (PeerNinja) [53], Ghana (Game-based SMS adherence) [54] or ongoing in South Africa (MAsakhane Siphucule Impilo Yethu; Xhosa for “Let’s empower each other and improve our health”/MASI) [55].

Fidelity to the intervention depends on the type of platform, for example, whether internet access is a requirement. General reading literacy appears to be important for all types of technology-enabled health interventions. For online or social media group or individual interventions, digital literacy, internet connectivity, and data are important. Further, clear instructions or training is required, especially if the intervention requires the adolescents to participate actively in various activities. Another consideration is whether the participants’ phones have such capabilities. The use of platforms that adolescents access frequently might improve fidelity [52]. However, competing demands of daily social messages and preference for using cell phones for communication with friends might interfere with intervention engagement. In an integrated review of technology-enabled interventions for ALHIV in the US, poor response rates (48% to 58%) to interactive SMS were identified as a concern. The studies that used interactive text messaging in this review also reported varied response rates from 20.5% [48] to 67.47% [40] out of the 86.4% [48] and 83.4% [40] of messages successfully delivered. For group sessions, appropriate scheduling is important. All the above indicate the importance of a needs analysis and the involvement of ALHIV when developing or adapting an intervention for a specific context [52].

## 5. Conclusions

There is a need to ensure that technology-enabled interventions have a strong theoretical base and that they can be integrated into existing healthcare or community services in order to promote sustainability and scale-up. Larger studies with rigorous evaluation designs are needed to determine the effects and effectiveness of technology-enabled interventions on the health outcomes of ALHIV in LMICs. Such studies should also report on which components of the intervention are most effective and which technology designs or platforms are acceptable and feasible for end-users (ALHIV) as well as health workers.

## Figures and Tables

**Figure 1 ijerph-20-02464-f001:**
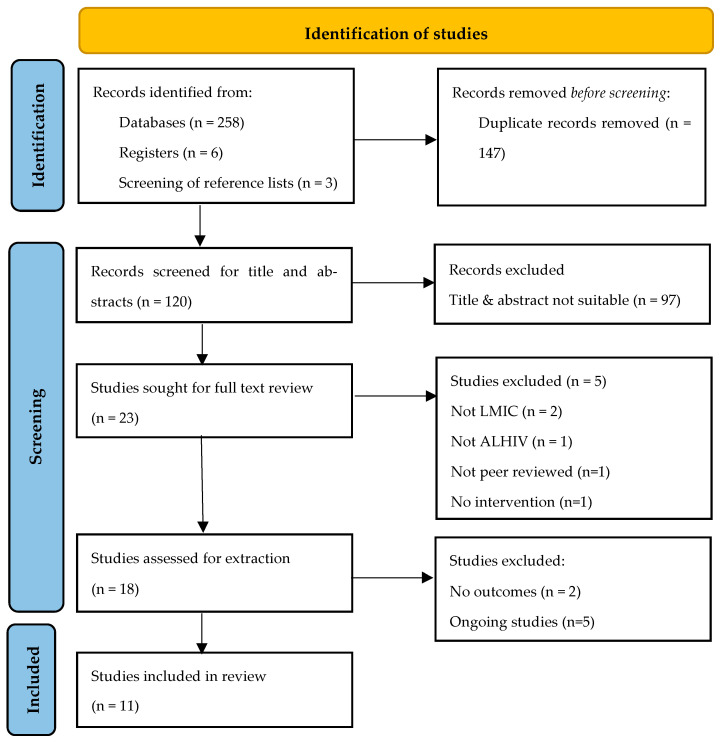
PRISMA diagram.

**Table 1 ijerph-20-02464-t001:** PICOT- criteria for inclusion of studies.

Component	Criteria
Patient/Population	Adolescents Living with HIV Aged 10–19 Years.
Intervention	Technology-enabled health interventions aimed at delivering or supporting health care directly to ALHIV
Comparisons	With or without comparison
Outcomes	Primary outcomes: health-related individual outcomes as specified by each study e.g., health/risk behaviours, self-management behaviours, self-efficacy, adherence, retention in care, viral suppression, quality of life, mental health or well-being. Secondary outcomes: process outcomes e.g., acceptability, feasibility, fidelity
Time	2010–2022
Other considerations	English language Low- and middle-income countries

**Table 2 ijerph-20-02464-t002:** Characteristics of included studies.

Authors, Year	Country	Study Design	Type of Data	Study Population	Total Number of Participants
Abiodun et al., 2021 [40]	Nigeria	RCT	Quantitative	ALHIV aged 15–19 years. The mean age was 16.61 (+/− 1.38) years. The study had 101 (48.33%) female and 108 (51.67%) male participants.	N = 209 (intervention group = 105; control group = 104)
Chory et al., 2022 [41,42]	Kenya	Non-RCT pre-post	Quantitative & qualitative	ALHIV aged 10–19 years. The mean age was 15.4 years, and the majority (56.7%) were female.	N = 30 (each group had 15 study participants at baseline; one group was 9–14-year-olds and the other 15–19-year-olds)
Dulli et al., 2018 [43]	Nigeria	Non-RCT pre-post	Quantitative & qualitative	ALHIV aged between 15–19 years old, 22 females, 19 males, and a mean age of 17 years.	N = 41 (5 support groups of 8–10 individuals)
Dulli et al., 2020 [44]	Nigeria	RCT	Qualitative & qualitative	Most participants were female 87.7% (306/349) and the mean age was 21 years (SD 2.3).	N = 349 (intervention group = 177 (50.6%); control group = 172 (49.1%)
Hacking et al., 2019 [45]	South Africa	Non-RCT matched controls	Quantitative & qualitative	HIV-positive youths, 95% female virtual mentees with a median age of 20 years 5 months; 91% female matched controls cohort with a median age of 22 years 7 months.	Virtual mentees cohort (N = 40) Matched controls cohort (N = 70)
Henwood et al., 2016 [46]	South Africa	Qualitative	Qualitative	HIV-positive youths between the ages 14–28 (59% between 23 and 25 years, 63% female).	60 club members completed the questionnaire, and 12 participated in the focus groups.
Ivanova et al., 2019 [47]	Kenya	Non-RCT pre-post	Quantitative	HIV-positive youths aged 15–24 years (36 were male and 45 were female). Mean age 18.4 years (SD = 2.8); range 15 to 25 years.	N = 90
Linnemayr et al., 2017 [48]	Uganda	RCT	Quantitative	HIV-positive adolescents and young adults aged 15 to 22 years. The mean was age 18 years; 61% were female.	N = 110 (message-only group), N = 110 (message and response group) N = 112 (control group)
McCarthy et al., 2020 [49]	Uganda	RCT	Quantitative & qualitative	HIV-positive youths 15–24 years.	N = 40 (treatment 1) N = 56 (treatment 2) N = 59 (control)
Sanchez et al., 2021 [50]	Guatemala	RCT	Quantitative	Age range of 6 to 12 years old (49.1%), and 13 to 24 years old (50.9%).	N = 72 (intervention group) N = 71 (control group)
Stankievich et al., 2018 [51]	Argentina	Non-RCT sequential design (pre-post)	Quantitative	Children and young people living with HIV; mean age 7.2± 6.1 years (range: 6–25); 11(50%) < 18 years; 15 (68%) females.	N = 25

**Table 3 ijerph-20-02464-t003:** Quality assessment.

Authors, Year	Study Design	Quality Assessment Score	Grading
Abiodun et al., 2021 [40]	RCT	12/12 (1)	Strong
Chory et al., 2022 [41,42]	Non-RCT pre-post	8/12 (0.66)	Moderately strong
Dulli et al., 2018 [43]	Non-RCT pre-post	6/12 (0.5)	Weak
Dulli et al., 2020 [44]	RCT	9/12 (0.75)	Moderately strong
Hacking et al., 2019 [45]	Non-RCT matched controls	10/12 (0.83)	Strong
Henwood et al., 2016 [46]	Qualitative	7/7 (1.00)	Strong
Ivanova et al., 2019 [47]	Non-RCT pre-post	5/7 (0.71)	Moderately strong
Linnemayr et al., 2017 [48]	RCT	5/7 (0.71)	Moderately strong
McCarthy et al., 2020 [49]	RCT	6/7 (0.85)	Strong
Sanchez et al., 2021 [50]	RCT	5/7 (0.71)	Moderately strong
Stankievich et al., 2018 [51]	Non-RCT pre-post (sequential design)	5/7 (0.71)	Moderately strong

**Table 4 ijerph-20-02464-t004:** Characteristics of technology-enabled interventions.

Authors, Year	Country	Name of the Intervention	Technology Design	Delivery Platform	Brief Description
Abiodun et al., 2021 [40]	Nigeria	STARTA Trial	Interactive individual	SMS	Duration 12 months Participants received daily ART adherence reminder SMS and were required to reply to their daily messages as soon as possible.
Chory et al., 2022 [41,42]	Kenya	A Mobile Intervention to Support Mental Health and Adherence Among Adolescents Living with HIV	Interactive groups	WhatsApp	Duration: 6 months Weekly in-person meetings to discuss topics e.g., adherence, disclosure etc. Informal WhatsApp communication is encouraged. The counsellor sends direct messages every other week.
Dulli et al., 2018 [43]	Nigeria	SMART (Social Media to promote Adherence and Retention in Treatment) Connections	Interactive groups	Facebook	Duration: 6 months Secret Facebook groups with safe space, trained adult facilitator, social activities (riddles, puzzles), interactive polls and facilitated discussions, word of the week, and key messages.
Dulli et al., 2020 [44]	Nigeria	SMART (Social Media to promote Adherence and Retention in Treatment) Connections	Interactive groups	Facebook	Duration: 22 weeks Same as for Dulli et al., 2018
Hacking et al., 2019 [45]	South Africa	The Virtual Mentors Program	Interactive individual	SMS, call, or WhatsApp	Duration: 14 months The mentors chatted with the mentees who responded to the messages concerning their families, and social activities, and invited them to the youth adherence clubs.
Henwood et al., 2016 [46]	South Africa	A virtual support group for HIV-positive youth	Interactive groups	Mxit	Duration: 13 months An adherence counsellor moderated the chat room for one hour on weekday afternoons and sent out a short message service to all registered chat room users to alert them when the counsellor joined the chat room. The counsellor-initiated conversations and promote interaction among users.
Ivanova et al., 2019 [47]	Kenya	ELIMIKA—digital peer support for improving ART adherence	Interactive groups	Web-based	Duration: 12 months Participants took part in blog post discussions with each other, project coordinators, and health care providers.
Linnemayr et al., 2017 [48]	Uganda	Text Messaging for Improving Antiretroviral Therapy Adherence	Interactive and non-interactive, individual (one-way vs. two-way vs. control)	SMS	Duration: 48 weeks Every Sunday at 9 AM, the program manager dispatched text messages to both intervention groups. Participants in the in the second group could respond to the messages.
McCarthy et al., 2020 [49]	Uganda	SITA (SMS as an Incentive To Adhere)	Non-interactive individual	SMS	Duration: 18 months Weekly messages were sent informing participants about their adherence level.
Sanchez et al., 2021 [50]	Guatemala	A text message intervention in an HIV paediatric clinic	Non-interactive individual	SMS	Duration: 6 months Frontline SMS was used to send SMS messages to the entire intervention group at the same time of the day. Messages were designed to improve each of the areas measured in the adherence questionnaire.
Stankievich et al., 2018 [51]	Argentina	Mobile Communication Devices as a Tool to Improve Adherence to Antiretroviral Treatment in HIV-Infected Children and Young Adults in Argentina	Interactive individual	Text messages sent via phone or Facebook	Duration: 32 weeks. A text message was sent through the application selected by the participant. A generic mobile message is sent twice a month. Patients agreed to answer the message to verify that the contact had been received.

**Table 5 ijerph-20-02464-t005:** Summary of primary outcomes.

Author, Year	Linkage to Care	Adherence	Viral Load	Retention in Care	HIV Knowledge	Social Support	Self-Efficacy for Adherence	Mental Health	Stigma	Behavioural Health
Abiodun et al., 2021 [40]	-	No	No	-	-	-	-	-	-	-
Chory et al., 2022 [41,42]	-	No	-	-	-	-	-	No	No	No
Dulli 2020 [44]	-	No	-	No	Yes	No	-	No	No	-
Hacking 2019 [45]	Yes	-	No	No	-	-	-	-	-	-
Ivanova et al., 2019 [47]	-	No	-	-	No	-	No	-	-	-
Linnemayr 2017 [48]	-	No	-	-	-	-	-	-	-	-
McCarthy 2020 [49]	-	No	-	-	-	-	-	-	-	-
Sanchez 2021 [50]	-	Yes	-	-	-	-	-	-	-	-
Steinkievich et al., 2018 [51]	-	Yes	Yes	-	-	-	-	-	-	-

Key: Yes = significance; No = non-significant.

**Table 6 ijerph-20-02464-t006:** Summary of secondary outcomes.

Study ID	Acceptability	Feasibility	Fidelity
Abiodun et al., 2021 [40]	High	High	Not reported
Chory et al., 2022 [41,42]	High	High	Not reported
Dulli et al., 2018 [43]	High	High	High
Dulli et al., 2020 [44]	High	High	High
Hacking et al., 2019 [45]	High	High	Not reported
Henwood et al., 2016 [46]	High	High	Low
Ivanova et al., 2019 [47]	High	High	Not reported
McCarthy et al., 2020 [49]	High	High	High
Sanchez et al., 2021 [50]	High	Not reported	Not reported
Steinkievich et al., 2018 [51]	High	High	Not reported

## Data Availability

The data that support the findings of this study are available from the corresponding author.

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
