# Peer review of "Effectiveness, Acceptability and Feasibility of Technology-Enabled Health Interventions for Adolescents Living with HIV in Low- and Middle-Income Countries: A Systematic Review"

_ijerph, 2023, doi:10.3390/ijerph20032464_

Round 1
Reviewer 1 Report
No comments
Author Response
Thank you for taking the time to review the manuscript. There are no comments to address.
Kind regards
Talitha Crowley
Reviewer 2 Report
1. According to world bank data there are 56 low and middle-income countries. This study included only 7 countries. Considering some more countries might be a better choice. Also, there is no inclusion of any Asian low- and middle-income countries, if included the study will be more inclusive.
2. Page no. 4 & 5, para 2.3, will there be any increase in the screened articles if HIV specific databases like Cochrane Library, Embase be included in the study.
3. Page no. 5, para 2.4 (Study selection), reference for the Covidence software is not provided.
4. Please include links/references for the used databases in the “2.3 Information and search strategy” under “2. Materials and Methods” section.
Author Response
Thank you for the comments. See below our response:
|
1. We selected all the available studies conducted in LMIC. 2. We do not believe that there would have been an increase in the number of included studies as our search strategy was comprehensive including 7 databases, Google Scholar, clinical trials registers etc. 3. Reference added and reference numbers revised. 4. It is not standard practice to include links or references for databases as these are standard databases. |

Reviewer 3 Report
Particularly complex study, with appropriate design, necessary to understand the elements through which adherence can be increased in adolescents living with HIV Perhaps it would be good to emphasize, during discussions, what were the results of similar studies from developed countries (USA)
Author Response
Thank you for the comment.
We have added information to the discussion section

Round 2
Reviewer 2 Report
The revised manuscript may be accepted.